# Molecular Links Between Metabolism and Mental Health: Integrative Pathways from GDF15-Mediated Stress Signaling to Brain Energy Homeostasis

**DOI:** 10.3390/ijms26157611

**Published:** 2025-08-06

**Authors:** Minju Seo, Seung Yeon Pyeon, Man S. Kim

**Affiliations:** 1Translational-Transdisciplinary Research Center, Clinical Research Institute, Kyung Hee University Hospital at Gangdong, Kyung Hee University School of Medicine, Seoul 05278, Republic of Korea; flora4713@khu.ac.kr; 2Department of Medicine, Kyung Hee University School of Medicine, Seoul 02453, Republic of Korea; 3Department of Obstetrics and Gynecology, Kyung Hee University Hospital at Gangdong, Kyung Hee University School of Medicine, Seoul 05278, Republic of Korea; pyun0522@khu.ac.kr

**Keywords:** metabolism, mental health, mitochondria, GDF15, gut–brain axis, precision medicine, biomarkers, depression, anxiety, stress

## Abstract

The relationship between metabolic dysfunction and mental health disorders is complex and has received increasing attention. This review integrates current research to explore how stress-related growth differentiation factor 15 (GDF15) signaling, ceramides derived from gut microbiota, and mitochondrial dysfunction in the brain interact to influence both metabolic and psychiatric conditions. Evidence suggests that these pathways converge to regulate brain energy homeostasis through feedback mechanisms involving the autonomic nervous system and the hypothalamic–pituitary–adrenal axis. GDF15 emerges as a key stress-responsive biomarker that links peripheral metabolism with brainstem GDNF family receptor alpha-like (GFRAL)-mediated anxiety circuits. Meanwhile, ceramides impair hippocampal mitochondrial function via membrane incorporation and disruption of the respiratory chain. These disruptions may contribute to sustained pathological states such as depression, anxiety, and cognitive dysfunction. Although direct mechanistic data are limited, integrating these pathways provides a conceptual framework for understanding metabolic–psychiatric comorbidities. Furthermore, differences in age, sex, and genetics may influence these systems, highlighting the need for personalized interventions. Targeting mitochondrial function, GDF15-GFRAL signaling, and gut microbiota composition may offer new therapeutic strategies. This integrative perspective helps conceptualize how metabolic and psychiatric mechanisms interact for understanding the pathophysiology of metabolic and psychiatric comorbidities and highlights therapeutic targets for precision medicine.

## 1. Introduction

The connection between metabolic disorders and mental health issues goes beyond a simple overlap. It reveals fundamental links that surpass traditional medical boundaries. The prevalence of metabolic–psychiatric comorbidities has reached alarming proportions, with substantial evidence demonstrating bidirectional relationships between these conditions [1,2]. This bidirectional relationship suggests shared underlying mechanisms that transcend conventional organ system boundaries.

Recent developments in molecular biology, systems neuroscience, and metabolomics have shown that changes in metabolism can directly influence brain function and behavior. Psychological stress can alter metabolism through neural and hormonal effects, creating a complex web of interactions that challenge traditional approaches to treating these conditions separately [3,4,5]. The emergence of precision medicine has highlighted the need for integrated therapeutic strategies that address both metabolic and psychiatric components simultaneously.

Despite significant advances in understanding individual pathways involved in metabolic and psychiatric disorders, a comprehensive framework that explains their interconnection has remained elusive. Traditional biomedical approaches have largely focused on single-system perspectives—either examining metabolic dysfunction in isolation or treating psychiatric symptoms without considering metabolic influences. This compartmentalized approach has contributed to suboptimal treatment outcomes and the persistence of treatment-resistant cases in both domains.

The complexity of metabolic–psychiatric interactions is further compounded by substantial individual differences in presentation, severity, and treatment response. These variations appear to be influenced by multiple factors including biological sex, age, genetic background, and environmental exposures. Women show higher prevalence of stress-related disorders compared to men, while also exhibiting distinct metabolic responses to stress that may contribute to differential disease susceptibility [6]. Similarly, aging processes affect both metabolic function and mental health through shared pathways involving cellular energy production and stress adaptation mechanisms [7,8,9].

This review explores how three main pathways influence metabolism and brain function: (1) stress-induced growth differentiation factor 15 (GDF15) signaling from adipose tissue lipolysis, (2) dysregulation of the gut–brain axis involving microbiota-derived metabolites, particularly ceramides, and (3) mitochondrial dysfunction in the brain that affects energy homeostasis. These pathways do not operate in isolation; they interact dynamically and create feedback loops that can amplify or attenuate metabolic and psychiatric symptoms based on individual factors such as age, sex, genetics, and environmental exposures.

The selection of these three specific pathways is based on emerging evidence that positions them as critical mediators of metabolic–psychiatric integration. GDF15 has recently been identified as a stress-responsive biomarker that directly links peripheral metabolic status with central anxiety circuits through brainstem GDNF family receptor alpha-like (GFRAL) receptors [10,11]. Ceramides represent a unique class of gut microbiota-derived metabolites that can cross the blood–brain barrier and directly impair hippocampal mitochondrial function [12,13]. Mitochondrial dysfunction serves as both a consequence and driver of metabolic–psychiatric pathology, creating self-perpetuating cycles of cellular energy failure and behavioral dysfunction [14,15].

Understanding these interconnected pathways is clinically important because metabolic–psychiatric comorbidities represent significant public health challenges globally. Current treatment approaches that address metabolic and psychiatric conditions separately often yield incomplete therapeutic responses and high relapse rates [1,2]. An integrative understanding of these interactions may enable the development of more effective, personalized interventions that target the root causes of comorbidity rather than treating symptoms in isolation.

The framework presented in this review aims to bridge the gap between basic research discoveries and clinical translation by providing a mechanistic understanding of how peripheral metabolic signals influence central nervous system function and vice versa. This bidirectional communication model may provide new opportunities for early detection, prevention, and treatment of metabolic–psychiatric comorbidities through targeted interventions that address pathway dysfunction at multiple levels simultaneously.

## 2. Integrative Model: Convergent Pathways Linking Metabolism and Mental Health

### 2.1. Conceptual Framework

The link between metabolism and mental health can be understood through three connected pathways that focus on the brain energy balance and include two-way feedback systems as illustrated in Figure 1.

#### 2.1.1. Primary Pathway: Stress-Induced GDF15 Signaling

Both acute and chronic stress activate hormonal responses that trigger the hypothalamic–pituitary–adrenal (HPA) axis. This results in the release of catecholamines and glucocorticoids. These stress hormones cause adipose tissue lipolysis, which serves as energy mobilization and a signal that prompts GDF15 production from adipose tissue macrophages. GDF15 then acts on brainstem GFRAL receptors to regulate energy balance and anxiety-like behaviors [10,16].

#### 2.1.2. Secondary Pathway: Gut–Brain Axis Dysregulation

Stress-related changes in the gut microbiota lead to increased levels of bioactive metabolites, especially ceramides. These ceramides enter the bloodstream and cross the blood–brain barrier. Ceramides disrupt mitochondrial function in the hippocampus by altering membrane fluidity, inhibiting respiratory chain complexes, and promoting mitochondrial fission. This ultimately contributes to depression [13,17].

#### 2.1.3. Tertiary Pathway: Central Mitochondrial Dysfunction

Brain areas with high energy needs, especially the prefrontal cortex and hippocampus, are particularly vulnerable to energy shortages. Chronic stress causes excessive mitochondrial fission and hinders recycling of damaged mitochondria, resulting in the development of dysfunctional mitochondria. This energy imbalance leads to cycles where damaged neurons become more sensitive to metabolic changes from the body [3,18,19].

### 2.2. Pathway Integration and Bidirectional Feedback Mechanisms

The interconnected nature and bidirectional feedback of these pathways play key roles in shaping the risk of metabolic and psychiatric comorbidities. Mechanistically, these pathway interactions occur through several means: (1) metabolic burden amplification, stress-related GDF15 signaling can overload already weakened mitochondrial systems impacted by ceramide accumulation; (2) shared vulnerability nodes, brain regions with high metabolic demands (such as the prefrontal cortex and hippocampus) work as convergence points where multiple pathways intersect; and (3) compensatory mechanisms, individuals with strong mitochondrial function might resist both GDF15-related anxiety signals and ceramide-driven dysfunction [20,21].

Dysfunction of the central nervous system creates reciprocal feedback loops, called bidirectional feedback loops, that maintain metabolic disorders in the body. Mitochondrial dysfunction caused by chronic stress in emotional brain regions can lead to ongoing HPA axis overactivity, resulting in constant catecholamine release and sustained GDF15 levels. Additionally, a compromised brain energy balance may change the output of the autonomic nervous system, such as vagal tone regulation, affecting the gut microbiota and further aggravating ceramide-related pathways [22,23]. The complex interplay between inflammatory processes, including cell-mediated immune activation and oxidative stress pathways, further contributes to the maintenance of these pathological cycles [24].

The interaction of these pathways helps explain several clinical observations: (1) the common co-occurrence of metabolic and psychiatric disorders, (2) the success of treatments that target multiple systems simultaneously, and (3) the wide variety of individual responses to treatment [1,2].

To provide a structured overview of these interconnected pathways and their molecular mediators, Table 1 summarizes key biomarkers that will be explored in detail in the following sections, highlighting their links to brain energy homeostasis, mental health outcomes, and clinical applications.

## 3. Peripheral Stress Signaling: The GDF15-GFRAL Pathway

### 3.1. Stress-Induced Hormonal Cascades and Metabolic Signaling

The body reacts to psychological stress by activating the neuroendocrine systems that mobilize energy resources and improve survival responses. This process begins when the hypothalamus releases corticotropin-releasing hormone, triggering the release of adrenocorticotropic hormone, which causes the adrenal glands to release glucocorticoids and catecholamines into the blood [3,25].

Catecholamine signaling during stress does not just help with immediate energy use. It also establishes long-lasting molecular communication between the peripheral tissues and brain. Catecholamines bind to β-adrenergic receptors on adipocytes, activating adenylyl cyclase and increasing intracellular cyclic AMP levels. This triggers lipolysis, the breakdown of triglycerides into free fatty acids and glycerol, which serve both as an energy substrate and signaling event [10].

Stress-induced lipolysis is not merely a metabolic response. It initiates the production of GDF15, a cytokine that responds to stress. GDF15 acts as a messenger between metabolism in the body and function in the brain. Studies using both restraint stress and pharmacological β3-adrenergic receptor stimulation have demonstrated that lipolysis is both necessary and sufficient for stress-induced GDF15 elevation, establishing a direct mechanistic link between psychological stress, peripheral metabolism, and brain signaling [10,16].

### 3.2. GDF15: A Dynamic Biomarker of Energetic Stress

Recent breakthroughs have established GDF15 as a useful biomarker for psychosocial stress in both blood and saliva. Controlled laboratory studies have shown that socio-evaluative stress quickly increases both GDF15 and lactate levels, indicating the activation of energetic and reductive stress pathways [11]. Importantly, salivary GDF15 shows a strong circadian wakening response. It peaks upon waking up and then declines by 42–92% within 30–45 min, similar to cortisol responses [12].

Mitochondrial stress in muscle tissues can trigger GDF15 release into the bloodstream. This release influences feeding behavior and anxiety through GFRAL-dependent pathways. The interaction between muscle and brain adds another dimension to the relationship between metabolism and mental health, with sex-specific differences observed in anxiety responses [16]. Additionally, GDF15 plays a role during pregnancy. Maternal mitochondrial health markers, including GDF15, are associated with prenatal stress and pregnancy outcomes [26].

### 3.3. Mechanistic Insights: From Lipolysis to GDF15 Production

GDF15 production during stress-induced lipolysis mainly occurs in M2-like macrophages within adipose tissue rather than adipocytes. Free fatty acids released during lipolysis activate peroxisome proliferator-activated receptor γ (PPARγ) in these macrophages. This leads to an increase in GDF15 transcription and secretion [10].

This mechanism represents a method of communication between the adipose tissue and brain. Metabolic stress in peripheral tissues creates signals that directly affect the central nervous system function. Genetic and pharmacological research indicate that blocking adipose triglyceride lipase (ATGL) or using the ATGL inhibitor, atglistatin, stops the increase in GDF15 during stress [10].

Bacterial infections are strong triggers of GDF15 expression. This broadens our understanding of how different stressors, including psychological, metabolic, and infectious factors, act on shared signaling pathways. Additionally, GDF15 may function as a general “stress integrator”, converting various types of physiological challenges into coordinated adaptive responses [27].

### 3.4. GDF15-GFRAL Signaling and Behavioral Regulation

The discovery of the GFRAL as a specific GDF15 receptor in 2017 is a major step forward in understanding the physiological functions of GDF15. Unlike other TGF-β family members that signal through serine/threonine kinase receptors, GDF15 signals through GFRAL, which works with the RET receptor, tyrosine kinase, to activate various intracellular pathways, including MAPK, PI3K/Akt, and PLC-γ signaling [28,29,30].

GFRAL is expressed only in certain brainstem areas, mainly in the postrema and nucleus of the solitary tract. This limited expression provides important clues about the mechanism of action of GDF15 [28]. These areas are shielded by an incomplete blood–brain barrier, allowing them to sense circulating factors directly and connect peripheral metabolic information with central nervous system function. Notably, these brainstem nuclei control feeding behavior, energy expenditure, autonomic functions, and emotional responses.

GDF15-GFRAL signaling plays a key role in stress-induced anxiety-like behaviors. Mice without GFRAL have normal HPA axis activation when faced with restraint stress or adrenaline but do not show typical anxiety-like behaviors. This suggests that GDF15-GFRAL signaling specifically influences behavior rather than neuroendocrine reactions to stress [10,16].

This specific behavioral response involves the activation of cholecystokinin-expressing neurons in the parabrachial nucleus, which connects to limbic structures, such as the amygdala and bed nucleus of the stria terminalis [31]. These brainstem–limb circuits create a pathway that allows peripheral metabolic signals to directly affect emotional processing and stress-related behaviors.

### 3.5. Clinical Implications and Therapeutic Potential

The discovery of GDF15 as a key mediator of stress–metabolism interactions has important clinical implications. Measuring GDF15 in the saliva makes it an appealing non-invasive biomarker for tracking stress reactivity and treatment [11,12]. Moreover, the role of GFRAL in influencing anxiety-like behaviors indicates the possibility of targeting this pathway for new anxiety treatments. However, the metabolic roles of GDF15 must be considered because chronic increases in levels have been linked to weight loss and metabolic issues in some cases [32,33].

## 4. Central Energy Metabolism and Mitochondrial Function

### 4.1. Brain Energy Requirements and Neuronal Vulnerability

The brain consumes 20% of the total body oxygen and 25% of the glucose for high-energy demands, despite comprising only 2% of the body weight. This reflects the ongoing ATP requirement for maintaining ionic gradients, supporting synaptic transmission, and healthy cells. Such high metabolic activity makes certain brain areas particularly susceptible to energy shortages, which has significant implications for mental health disorders [3,4,34].

Among the brain cells, neurons are the most vulnerable to energy disruptions because they have limited energy storage and constant metabolic needs. Sodium potassium ATPase alone uses approximately 70% of neuronal ATP, highlighting the importance of efficient energy production for essential neuronal functions [3]. Differences in energy use within the brain indicate functional specialization, with the prefrontal cortex and hippocampus being key areas for mood regulation and thinking, showing particularly high metabolic rates and increased susceptibility to energy shortages [4].

Alcohol use disorders are linked to metabolic changes that affect mitochondrial function. Certain metabolites, such as succinic acid and N6-acetyl-lysine, act as biomarkers that connect mitochondrial dysfunction with emotional symptoms [35]. Additionally, neuropsychiatric issues stemming from COVID-19 have been linked to poorer long-term outcomes, possibly involving pathways related to mitochondrial dysfunction [36].

### 4.2. Mitochondrial Dynamics and Stress Vulnerability

Mitochondria in neurons constantly undergo fusion and fission, which help maintain their quality, adapt to changing energy needs, and ensure proper distribution of mitochondria throughout the neuron. This regulation is crucial for neurons due to their complex structures and varied energy demands [3,37].

Mitochondrial fusion, which involves mitofusin 1 (MFN1), mitofusin 2 (MFN2), and optic atrophy 1 (OPA1), allows mitochondria to share contents and form long networks that efficiently spread energy throughout the cell. This process becomes critical during acute stress, because neurons require rapid increases in ATP production to handle stress responses [3,4].

In contrast, mitochondrial fission, mainly controlled by dynamin-related protein 1 (DRP1), helps to remove damaged parts and transport mitochondria to distant areas within neurons. However, chronic stress can lead to excessive DRP1 activation, which causes harmful mitochondrial fragmentation and dysfunction, rendering neurons more vulnerable to additional stress [18].

Chronic stress-induced mitochondrial fission in the medial prefrontal cortex directly leads to behavioral issues. Drug inhibition of DRP1 improves stress-related behavioral problems. These findings suggest a link between changes in mitochondrial activity and stress-related mental health symptoms, thus offering a potential treatment target [18]. Furthermore, research involving antibiotic-induced gut dysbiosis models indicated that mitochondrial dysfunction in the hippocampus and prefrontal cortex contributes to behaviors similar to those of autism, which can be improved by focusing on mitochondrial function [38] (Figure 2).

### 4.3. Mitophagy and Quality Control in Mental Health

Mitophagy, the selective breaking down of damaged mitochondria, is essential for maintaining neuronal health. This process becomes even more crucial during stress when mitochondrial damage exceeds the repairing capacity of cells. Therefore, removing dysfunctional mitochondria is necessary [14].

The PINK1/Parkin pathway is the most studied mitophagy mechanism. When the mitochondrial membrane potential decreases, PINK1 accumulates and attracts Parkin. This recruitment leads to targeted degradation of mitochondria. Disruptions in this pathway play a major role in depression, particularly those involving the mitophagy receptor NIX (BNIP3L) [14].

Lu et al. showed that tumor necrosis factor-α (TNF-α), a pro-inflammatory cytokine that increases in depression, triggers the degradation of the mitophagy receptor, NIX. This degradation causes accumulation of damaged mitochondria, leading to synaptic defects and passive stress-coping behaviors in mice. The study indicated that TNF-α disrupted mitophagy in the medial prefrontal cortex (mPFC) by promoting the breakdown of NIX, an outer mitochondrial membrane protein. The loss of NIX-mediated mitophagy results in more damaged mitochondria, which contribute to synaptic issues and abnormal behavior [14].

Importantly, eliminating NIX from excitatory neurons in the mPFC causes passive coping behaviors in response to stress, resembling the symptoms of depression. Additionally, increasing NIX levels in the mPFC reversed the behavioral problems caused by TNF-α. Ketamine, a fast-acting antidepressant, partly produces its effects by activating NIX-mediated mitophagy. Furthermore, low levels of NIX were found in the blood of patients with major depressive disorder and mPFC tissues of animal models. Infliximab, a drug that blocks TNF-α, reduced both chronic stress and inflammation-related behavioral issues by restoring NIX levels [14].

The importance of mitophagy as a potential treatment is highlighted by the finding that various natural compounds, such as oligosaccharides from Morinda officinalis and gypenosides from Gynostemma pentaphyllum, improve depression-like behavior by enhancing mitophagy. This suggests that promoting mitophagy is a promising treatment approach [39,40]. Additionally, agomelatine reduces depression-like behaviors by decreasing oxidative stress in the hippocampus and restoring mitochondrial function [41]. Diosmetin, a flavonoid compound, similarly improves cognitive and behavioral deficits by enhancing mitochondrial function and reducing neuroinflammation in high-fat diet-induced rats [42].

### 4.4. Clinical Evidence for Mitochondrial Dysfunction in Mental Health

Mitochondrial dysfunction has been confirmed in mental health disorders. One important study used functional near-infrared spectroscopy during verbal fluency tasks to check the blood flow in the brains of first-episode, drug-naïve patients with major depressive disorders. The results showed a significant reduction in frontotemporal activation during cognitive tasks, which was correlated with lower serum levels of succinate dehydrogenase, a key mitochondrial enzyme. This study linked peripheral mitochondrial dysfunction to changes in brain activity patterns in depression [43].

Studies examining the relationship between mitochondrial DNA and lifestyle factors have revealed significant connections between mitochondrial genetic variants and psychiatric disorders. The interaction between mitochondrial DNA variations and lifestyle choices, such as smoking and drinking, substantially influences the risk of developing anxiety, depression, and self-harming behaviors [44]. Additionally, apolipoprotein E genotypes affect neuropsychiatric symptoms through their impact on mitochondrial metabolism and neuroinflammation pathways [45].

Transcriptomic analysis of the amygdala in patients with schizophrenia, bipolar disorder, and major depressive disorder revealed different metabolic pathways. Schizophrenia exhibits the downregulation of mitochondrial respiration pathways, whereas major depressive disorder displays the upregulation of energy metabolism pathways [15]. These findings suggest that different diagnoses may lead to specific changes in brain energy metabolism, which may inform treatment strategies.

### 4.5. Therapeutic Targeting of Mitochondrial Function

Recognizing mitochondrial dysfunction as a key aspect of many psychiatric disorders has led to new treatment methods that target mitochondrial health. Extremely low-frequency electromagnetic field therapy can improve the mitochondrial electron transport chain activity and reduce depressive behavior by activating the Sirt3-FoxO3a-SOD2 pathway [46].

Ketamine, a rapid-acting antidepressant, reverses the inhibition of the mitochondrial respiratory chain caused by chronic mild stress. Ketamine administration can restore the activity of complexes I, III, and IV in the cerebral cortex and cerebellum. This suggests that the antidepressant effects of ketamine may partly involve restoration of mitochondrial function [47]. This finding offers further insight into the unique therapeutic benefits of ketamine beyond blocking NMDA receptors.

Physical exercise is another effective method of improving mitochondrial function and mental health. Aerobic exercise can reduce psychiatric disorders and cognitive impairment by boosting mitochondrial function and neuroplasticity in post-traumatic stress disorder [48]. Exercise increases mitochondrial activity in the brain, improves resistance to complex I inhibitors, and increases the expression of genes involved in energy metabolism and BDNF regulation [49].

Nutritional approaches have the potential to protect mitochondrial function and support mental well-being. omega-3 fatty acids, antioxidants, and other nutrients can shield mitochondria and membrane lipids in neuronal circuits related to cognitive and emotional behaviors from oxidative damage [50,51]. The endocannabinoid system, which is influenced by dietary omega-3 fatty acids and lifestyle choices, plays a considerable role in mitochondrial function and mood regulation [50].

Other promising methods include natural products from traditional medicine. Several compounds have demonstrated antidepressant effects by modulating mitochondrial function and mitophagy pathways [52,53]. Additionally, oxidative stress impacts fatty acid and one-carbon metabolism, which may link psychiatric and cardiovascular diseases. This highlights the need for integrated metabolic treatment [54].

## 5. Gut–Brain Axis: Microbiota-Mediated Metabolic Signaling

### 5.1. Microbiota Composition and Metabolite Production

The gut microbiota, which is composed of trillions of microorganisms living in the gastrointestinal tract, plays a key role in regulating host metabolism and mental health. Factors, such as diet, lifestyle, stress, and genetics influence the composition of gut microbiota. This composition can affect many aspects of host physiology by producing bioactive metabolites [55,56].

Psychological stress changes the composition of the gut microbiota. This shift leads to the production of more metabolites that affect brain function. In research using unpredictable chronic mild stress (UCMS) as a mouse model for depression, mice show changes in behavior and biology. These changes can be transferred from UCMS donor mice to naïve recipient mice via fecal microbiota transplantation [55] and were linked to a drop in endocannabinoid (eCB) signaling, which resulted from lower levels of fatty acid precursors for eCB ligands.

Compounds from traditional medicines can influence gut–brain axis function by affecting intestinal bacteria and glycerophospholipid metabolism. Total flavone content of Abelmoschus manihot can be used to treat ulcerative colitis and depression. This is achieved by correcting imbalances in intestinal flora and reshaping glycerophospholipid metabolism, which inhibits M1 macrophage activation [57].

### 5.2. Ceramides: A Critical Link Between Gut Dysbiosis and Depression

Ceramides are important metabolites that connect gut microbiota imbalances to depression by directly affecting mitochondrial function in the brain. In a corticosterone-induced depression model, researchers showed that dysbiosis in the gut leads to high ceramide levels in the intestinal lumen. These ceramides enter the bloodstream and accumulate in the brain, especially in the hippocampus [13].

In hippocampal tissue, ceramides disrupt mitochondrial function in several ways: they (1) integrate into mitochondrial membranes, changing their fluidity and affecting the activity of respiratory chain complexes; (2) induce harmful mitochondrial fission; (3) inhibit mitophagy; and (4) encourage mitochondrial permeability transition, which causes cytochrome c release and cell death [13].

The role of gut microbiota in this process was confirmed through fecal microbiota transplantation experiments. When the microbiota from depressed mice was transferred to germ-free mice, both dysbiotic characteristics and depressive behaviors were transferred. Notably, treatment with probiotics, such as Bifidobacterium pseudolongum and Lactobacillus reuteri, restores gut microbiota balance, lowers ceramide levels, and reduces depressive-like behaviors [13].

Prenatal depression is also linked to changes in glycerophospholipid and sphingolipid metabolism driven by the gut microbiota. Transferring fecal microbiota from women with prenatal depression into germ-free mice causes depressive-like behaviors and specific issues in sphingolipid metabolism, particularly those related to ceramide types [17].

### 5.3. Therapeutic Targeting of the Gut–Brain Axis

The discovery of specific metabolic pathways connecting gut microbiota to brain function has created new treatment options. Certain combinations of probiotics can normalize the gut microbiota composition, lower ceramide levels, and reduce depressive behaviors. This approach to probiotic therapy is based on a clear understanding of the specific metabolic pathways [13,17].

The potential of synbiotic interventions that combine probiotics and prebiotics to treat neurological disorders has been emphasized. These interventions may improve cognitive function and lower inflammation [58]. The possibility of targeting the microbiota-gut–brain axis includes various methods, such as dietary changes, fecal microbiota transplantation, and microbiome-focused treatments aimed at restoring healthy microbial balance and metabolic function.

## 6. Sex Differences, Age-Related Changes, and Genetic Modulation

### 6.1. Sexual Dimorphism in Metabolic–Psychiatric Connections

#### 6.1.1. Sex-Specific Stress Response Patterns

##### Acute vs. Chronic Stress Responses

Acute stress responses demonstrate pronounced sex differences at both behavioral and molecular levels. In controlled studies using acute restraint stress, males and females showed distinct patterns of recognition memory alterations and AMPA/NMDA receptor subunit changes, with females exhibiting increased anxiety-like behaviors while males primarily displayed cognitive deficits in temporal order recognition tasks [59]. The mechanisms underlying these sex-specific responses involve differential activation of stress-response circuitry, where females demonstrate greater negative affective responses to stress than males beginning in adolescence, despite showing comparatively lower peripheral physiological responses in the hypothalamic–pituitary–adrenal axis and autonomic nervous system reactivity [60].

These acute stress differences extend to chronic stress responses with distinct temporal patterns. Extended chronic restraint stress (42 days) revealed sex-specific impacts on instrumental learning, with stressed males showing increased response speed but lower learning efficiency, while stressed females demonstrated slower appetitive response speeds but higher learning efficiency overall [61]. Importantly, males exhibited significant changes in phosphorylated CREB levels and brain morphology during stress responses, whereas females showed no such response, suggesting fundamentally different neural adaptation mechanisms [62].

##### Hormonal and Neural Mechanisms

The HPA axis demonstrates clear sexual dimorphism in both basal activity and stress responsiveness. Among older adults, sex differences in HPA reactivity become more pronounced, with older unfit women showing significantly greater cortisol responses to psychological stress compared to both young women and older fit women [63]. These age-related sex differences suggest that hormonal and fitness factors interact to modulate stress responsiveness across the lifespan. Additionally, sex differences are evident in HPA axis activity in response to metabolic challenges, with males showing higher postprandial cortisol responses than females, independent of body composition or psychological variables [64].

#### 6.1.2. Metabolic–Psychiatric Comorbidity Patterns

##### Clinical Manifestations

Sex differences in metabolic–psychiatric comorbidities are evident across multiple disorders. In major depressive disorder patients with comorbid glucose metabolism abnormalities, significant sex differences emerge in dyslipidemia prevalence and risk factors, with distinct patterns of lipid metabolism correlates between males and females [65]. Large-scale population studies demonstrate that sex differences in psychiatric and metabolic comorbidity prevalence are observed among adults with ADHD, with effect modification by sex detected on both additive and multiplicative scales for associations with all comorbidities [66].

##### Underlying Mechanisms

The underlying mechanisms involve sex-specific associations between metabolic hormones, severe mental disorders, and antipsychotic treatment, indicating sex-dependent mechanisms in metabolic regulation [67]. Furthermore, first-episode drug-naive schizophrenia patients show sex differences in metabolic disorder patterns, with complex interactions between sex hormones affecting adiposity deposition and metabolic syndrome development [68].

#### 6.1.3. Mitochondrial Function and Sex-Specific Responses

Separate patterns occur in mitochondrial regulation between males and females with major depressive disorders, suggesting that different mechanisms are involved. Deleting SIRT1 in forebrain excitatory neurons leads to depression-like symptoms in males but not in females. This difference may be due to the regulation of mitochondrial growth and activity, as SIRT1 knockout results in lower mitochondrial density in the male prelimbic prefrontal cortex [69]. These findings show that SIRT1 in mPFC excitatory neurons are required for normal neuronal activity and synaptic transmission, and regulate depression-related behaviors in a sex-specific manner.

### 6.2. Age-Related Changes in Metabolism–Mental Health Connections

#### 6.2.1. Peripheral Stress Signaling and Aging

Aging significantly affects metabolic and mental health connections, including peripheral stress signaling to central mitochondrial function. GDF15 levels increase with age and are associated with frailty, inflammation, and slow recovery from illness in older adults. The increase in GDF15 may reflect ongoing low-grade energetic stress that can make older individuals more prone to mood disorders [7].

Age-related changes in HPA axis function demonstrate significant interactions with sex. Among women, aging is associated with greater HPA axis reactivity to psychological stress, but higher aerobic fitness among older women can attenuate these age-related changes as indicated by blunted cortisol responses [63]. These findings suggest that lifestyle interventions may be particularly important for maintaining metabolic–mental health connections in aging populations.

#### 6.2.2. Mitochondrial Function and Aging

Mitochondrial function declines with age due to several factors, including reduced biogenesis, poor quality control, and increased oxidative damage. Age-related changes in mitochondria increase the risk of stress-related behavioral problems, suggesting that aging increases vulnerability to developing metabolic and psychiatric disorders. Lack of adiponectin can accelerate brain aging through mitochondria-related neuroinflammation, highlighting the complicated relationship between metabolic hormones, aging, and brain function [8,9].

#### 6.2.3. Social and Environmental Aging Effects

The relationship between aging and metabolic–mental health connections shows clear sex-specific patterns influenced by social factors. Prospective cohort studies demonstrate that marital status impacts metabolic syndrome development differently in males and females, with divorced or widowed males at higher risk compared to single and married males, while married females show increased risk [70]. These findings suggest that social determinants of health interact with biological sex differences in ways that affect metabolic and mental health outcomes across the lifespan.

### 6.3. Genetic Modulation of Pathway Function

#### 6.3.1. Genetic Variants in Stress-Related Pathways

The significant individual differences in metabolism-related mental health partly originate from genetic variations that influence key pathways. Genome-wide association studies have identified specific genetic variants associated with anxiety and stress-related disorders, with variants in PDE4B showing significant associations (odds ratio = 0.89; 95% CI, 0.86–0.92) [71]. Importantly, these genetic effects demonstrate sex-specific patterns, with single-nucleotide polymorphism heritability of 28% and genetic signatures overlapping with psychiatric traits, educational outcomes, and obesity-related phenotypes [71].

##### Stress Resilience Genetics

Genetic variants affecting stress resilience demonstrate clear mechanistic pathways. Variations in neuropeptide Y (NPY) gene expression affect anxiety reduction and stress tolerance, with people carrying low-NPY variants showing heightened emotional responses to threatening stimuli and reduced pain tolerance [72]. These individuals demonstrate increased metabolic activity in brain regions involved in emotional processing and release less opioid neurotransmitter in response to stress, providing a molecular basis for individual differences in stress susceptibility.

#### 6.3.2. Mitochondrial Genetic Variants

Detailed analysis of single-nucleotide polymorphisms in genes responsible for maintaining mitochondrial DNA has shown strong links with the occurrence, onset, severity, and response to depression treatment. Large-scale genomic studies have revealed multiple mitochondrial SNPs linked to subjective well-being and psychiatric outcomes, with distinct patterns based on sex [73,74].

The interaction between mitochondrial DNA and lifestyle factors showed that smoking and drinking can influence the expression of mitochondrial genes, potentially affecting psychiatric disorders by altering mitochondrial respiratory chain activity and gene regulation [44]. These findings indicate that lifestyle changes can have different effects depending on an individual’s genetic makeup.

#### 6.3.3. Metabolic Syndrome Genetics

Genetic enrichment studies have identified pathogenic variants in genes associated with inborn errors of metabolism in psychiatric populations, suggesting shared genetic vulnerabilities between metabolic and psychiatric disorders [75]. These findings support the hypothesis that psychiatric populations are enriched for variants that affect fundamental metabolic processes, providing a genetic basis for the observed clinical comorbidities.

### 6.4. Environmental Modulation of Metabolic–Psychiatric Connections

#### 6.4.1. Socioeconomic Factors

##### Stress Hormone Regulation

Socioeconomic status demonstrates graded associations with metabolic–mental health pathways through multiple mechanisms. Lower SES is associated with higher basal levels of cortisol and catecholamines in a dose-dependent fashion, independent of race, with associations mediated by health practices and social factors [76]. These hormonal changes create a chronic stress state that affects both metabolic function and mental health outcomes.

##### Developmental Effects

Socioeconomic factors influence stress interpretations and physiological responses from adolescence onwards. Lower SES adolescents show greater threat interpretations during ambiguous situations and exhibit greater cardiovascular reactivity, with threat interpretations partially mediating the relationship between SES and stress reactivity [77]. General life events, rather than specific stressors, explain much of the relationship between low SES and maladaptive stress responses.

#### 6.4.2. Social and Work Environment

##### Work–Life Balance

Environmental stressors in occupational settings demonstrate complex interactions with socioeconomic status in affecting mental health. Work–life conflict shows differential effects on mental health depending on SES level, with higher SES individuals paradoxically showing greater sensitivity to work–life conflict effects, possibly due to higher aspirations and expectations [78]. Family demands emerge as critical factors that exacerbate work–life conflict and its mental health consequences.

##### Chronic Social Stress

Comprehensive research indicates that chronic social stress and low socioeconomic status are associated with metabolic syndrome development through multiple pathways [76]. The cumulative effects of socioeconomic stress on health and well-being are evident throughout the lifespan, affecting children, adolescents, and adults through mechanisms that may be targeted for treatment and prevention strategies.

#### 6.4.3. Epigenetic Environmental Effects

##### Gene–Environment Interactions

Environmental factors interact with the genome through epigenetic mechanisms that regulate gene expression and contribute to psychiatric disorder pathogenesis. Genetic variants in stress-response genes, such as FKBP5, show increased expression with age due to reduced DNA methylation, leading to glucocorticoid resistance and reduced coping behavior [71]. These findings demonstrate how environmental factors can create lasting changes in stress-response systems through molecular mechanisms.

### 6.5. Integrative Framework: Multi-Level Interactions

The evidence reviewed demonstrates that sex, age, genetics, and environment operate as interconnected factors rather than independent variables in shaping metabolic–psychiatric connections. Sex differences in acute and chronic stress responses [60,61] interact with age-related changes in HPA axis function [63] and are modulated by genetic variants affecting stress resilience [71,72] and environmental factors such as socioeconomic status [76,77].

This multi-level interaction framework has important implications for understanding individual differences in susceptibility to metabolic–psychiatric comorbidities and for developing personalized intervention strategies. The dynamic interplay between these factors creates both vulnerabilities and opportunities for therapeutic intervention across the lifespan.

## 7. Clinical Implications and Therapeutic Strategies

### 7.1. Novel Therapeutic Modalities

Several new therapeutic approaches that focus on the link between metabolism and mental health have been established. These methods operate through processes that are not typically linked to psychiatric treatment. Extremely low-frequency electromagnetic field therapy has shown promising results in early studies. This treatment increases the mitochondrial electron transport chain activity and improves depressive behavior by activating the Sirt3-FoxO3a-SOD2 pathway [46].

Exercise interventions are evidence-based methods that simultaneously target multiple pathways. Physical exercise can improve psychiatric disorders by enhancing the mitochondrial function in the hippocampus and promoting neuroplasticity. These benefits include improved mitochondrial activity, increased BDNF expression, and increased resistance to mitochondrial inhibitors [48,49]. Additionally, emerging evidence suggests that inflammatory cytokines, particularly those associated with appetite loss, represent important therapeutic targets in adolescent depression [79].

Natural products continue to produce promising compounds that target mitochondrial function and metabolism. Various compounds that improve cognitive impairment and depressive behavior by boosting mitochondrial function and reducing neuroinflammation have been identified [52,53]. Additionally, diosmetin improves cognitive impairments by addressing metabolic disorders, mitochondrial dysfunction, and neuroinflammation [42].

Nutritional interventions targeting the mitochondrial function are particularly promising. omega-3 fatty acids, antioxidants, vitamin B compounds, and magnesium can protect the mitochondria from oxidative damage and support neurotransmitter signaling [51,56]. These nutrients may help prevent mood disorders and enhance the effectiveness of current antidepressant efficacy. Furthermore, dietary inflammatory indices have been shown to correlate with postpartum depression risk, highlighting the importance of anti-inflammatory dietary approaches [80].

### 7.2. Precision Medicine Applications

Significant differences in how individuals function and respond to treatment require personal therapeutic methods that consider the unique weaknesses and strengths of each individual. Recent progress in metabolomic profiling has enabled the identification of individual metabolic patterns linked to psychiatric symptoms. Acylcarnitine profiles can measure energy capacity and predict treatment responses, particularly in individuals with certain mitochondrial genetic variants [1,81].

Acylcarnitine plays a role in energy processes that may contribute to the development of depression. Changes in short-chain acylcarnitine levels are related to the presence and severity of depression, particularly symptoms associated with energy imbalance [81]. These results suggest that cellular metabolism may be a crucial factor in the pathophysiology of depression and can be addressed through metabolic treatments.

### 7.3. Biomarker Development and Clinical Translation

The translation of metabolism–mental health research into clinical practice requires the development of accessible and reliable biomarkers to guide diagnosis and treatment decisions. Recent advances in multi-omic integration approaches have potential for improving the blood biomarkers of major depressive disorder. This demonstrates the potential of combining multiple biological measures to improve diagnostic accuracy [2].

Circulating mitochondrial DNA and related biomarkers have been identified as potential signs of mitochondrial dysfunction in psychiatric disorders. This is particularly true for older adults with mild cognitive impairment and remitted major depressive disorder [30]. These findings suggest that peripheral biomarkers of mitochondrial function offer valuable insights into brain health and treatment responses. Moreover, the mediating effects of pro-inflammatory cytokines in the association between depression, anxiety, and cardiometabolic disorders provide additional biomarker opportunities for understanding disease mechanisms and treatment targets [82]. These strategies are summarized in Table 2.

## 8. Future Research Directions and Clinical Translation

### 8.1. Advanced Biomarker Development

Translating metabolic and mental health research into clinical practice requires the development of accessible and reliable biomarkers to guide diagnoses and treatment decisions. Salivary GDF15 profiling is a particularly promising approach because it is non-invasive and can change over time [11,12]. Additionally, metabolomic profiling of treatment-resistant major depressive patients with suicidal thoughts shows distinct metabolic patterns that can guide treatment options [2].

An individual’s bioenergetic capacity, assessed using acylcarnitine profiles and other metabolic markers, may act as a potential source of resilience against neurodegenerative diseases and psychiatric disorders [1]. These findings indicate that metabolic resilience biomarkers can help to identify at-risk individuals and guide preventive measures.

### 8.2. Technology Integration and Digital Health

The complexity of metabolic and mental health interactions provides opportunities for technology-based assessment and intervention methods. Integrating wearable devices could allow ongoing monitoring of physiological factors related to metabolism and mental health connections. This includes heart rate variability measures that indicate vagal tone and autonomic function.

Recent progress in bioinformatics and machine learning has made it possible to analyze multi-omic datasets more effectively. This allowed us to identify complex patterns that connect genetic, metabolic, and clinical variables. These methods have the potential to create predictive models for treatment responses and personalized intervention strategies [1].

## 9. Limitations and Critical Considerations

### 9.1. Translational Challenges

Although preclinical research provided strong evidence for the pathways linking metabolism and mental health, several challenges need to be resolved for successful clinical application (Table 3). Differences in metabolism, stress responses, and brain structures between species may limit the application of rodent findings to humans. Human mental health disorders are far more complex than those that can be modeled in laboratory animals. Conditions, such as depression and anxiety, involve complicated interactions between social, psychological, and environmental factors [6,8].

The manner in which compounds that target metabolism and mental health pathways are absorbed and processed can substantially vary between controlled laboratory conditions and real-world clinical settings. The changing nature of the metabolic–mental health relationship creates obstacles for treatment. Key aspects, such as the right timing, treatment duration, and long-term effects of pathway changes, remain largely uncertain.

### 9.2. Methodological Considerations

Confounding variables in human studies on metabolism and mental health connections include diet, physical activity, sleep patterns, medication use, and other medical conditions. These factors can significantly affect metabolic parameters and mental health outcomes. This influence can either hide or exaggerate specific effects [2]. The changing nature of many biomarkers related to metabolism and mental health requires clear protocols for collecting samples and interpreting the results [11,12].

Concerns regarding reproducibility were highlighted in metabolomics and microbiome research. This was partly due to the differences in the analytical methods used and variations in the study populations. To establish reliable biomarkers, it is crucial to standardize methodologies and validate them across multiple independent groups [13,17].

### 9.3. Ethical and Safety Considerations

However, the long-term effects of interventions targeting basic metabolic pathways remain largely unknown. Changing GDF15 signaling or mitochondrial function over a long period can lead to unexpected problems that may not appear until after a long period of treatment [10,33]. Differences in individual responses to treatment raise ethical issues regarding fair access to personalized therapies [1,2].

Interventions are accompanied by risks, especially for preventive measures in high-risk individuals based on biomarker analysis. Therefore, the benefits of early intervention need to be weighed against the risks of unnecessary treatment [11,12].

## 10. Conclusions and Clinical Implications

This review provides a strong foundation for understanding how peripheral metabolism is linked to mental health through three main pathways: stress-related GDF15 signaling from fat tissue breakdown, gut–brain axis issues involving ceramide-related mitochondrial damage, and energy imbalances in the central nervous system with mutual feedback mechanisms. Together, these pathways offer a framework for understanding the connection between metabolic and psychiatric conditions, and for developing specific treatments.

Key scientific advances from this research include the following: (1) discovering GDF15 as a key marker connecting psychological stress with metabolic responses and brain activity through certain GFRAL pathways [10,11,12]; (2) identifying specific metabolites from gut bacteria, especially ceramides, that disrupt brain mitochondrial function by embedding in membranes and blocking the respiratory chain [13,17]; (3) describing feedback loops where central mitochondrial issues affect peripheral metabolism through autonomic and hormone systems [14,46]; (4) documenting differences in these pathways based on sex, age, and genetics, which could influence personalized treatment strategies [6,7,8,9]; and (5) confirming mitochondrial dysfunction as a common factor in various psychiatric disorders with metabolic elements [14,15,18].

Clinical translation opportunities extend across many healthcare areas. The presence of GDF15 in saliva provide opportunities for non-invasive stress monitoring and early intervention in both community and clinical environments [11,12]. Understanding ceramide-related gut–brain signaling fosters the development of precise probiotic therapies that target specific metabolic routes rather than using broad methods [13,17]. Recognizing mitochondrial dysfunction as a treatment target has led to new avenues for developing treatments, including specific antioxidants, energy boosters, and electromagnetic field therapy [46,52].

Personalized medicine applications can benefit from the significant individual differences in pathways and treatment responses. Multi-omics profiling can help identify individual weaknesses in pathways, and interventions based on biomarkers can improve treatment choices and monitoring [1,2,81]. Developing treatments that consider sex differences in metabolism and mental health connections may enhance outcomes for both men and women through tailored approaches [6,7,8,9]. Genetic profiling of mitochondrial function variations, stress response genes, and metabolic capacity can help select the best therapeutic strategies [44,73,74].

Public health implications of this study extend beyond individual treatment to include prevention and broader health strategies. Understanding how stress, metabolism, and mental health interact provides targets for interventions that can prevent the development of chronic conditions [2]. Identifying modifiable factors, such as gut microbiota, mitochondrial health, and stress responses, creates opportunities for lifestyle and dietary changes that can significantly impact public health [55,58]. Screening at the population level using easy-to-access markers, such as salivary GDF15 or circulating acylcarnitines, can help identify at-risk individuals early.

Evidence supporting pathways that connect metabolism and mental health signifies a major shift in our understanding of psychiatric disorders. Instead of seeing mental health issues as mere neurochemical imbalances, this study shows that signals from metabolism, gut bacterial activity, energy processes, and feedback loops play crucial roles in psychiatric symptoms. This broader understanding opens new avenues for creating more effective and personalized treatments that address the underlying causes of metabolic and psychiatric issues.

Implementing these insights clinically may provide better outcomes for millions of people suffering from the complex overlap of metabolic dysfunction and mental health disorders. With precision medicine guided by a solid understanding of mechanisms, the field can move beyond managing symptoms to targeted treatments that restore the integration between metabolism and mental health.

## Figures and Tables

**Figure 1 ijms-26-07611-f001:**
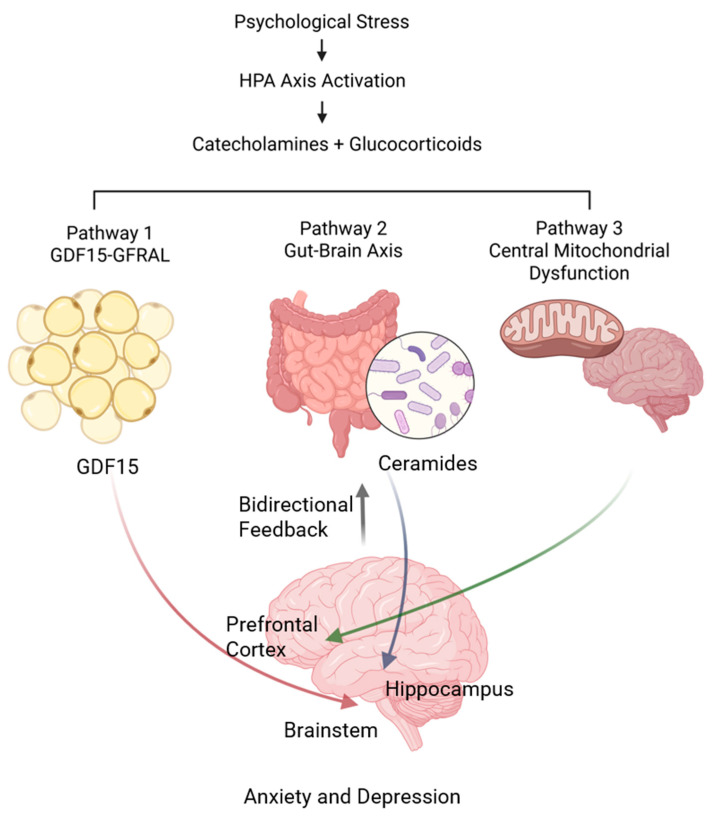
Integrative model of metabolism–mental health pathways. HPA, hypothalamic–pituitary–adrenal.

**Figure 2 ijms-26-07611-f002:**
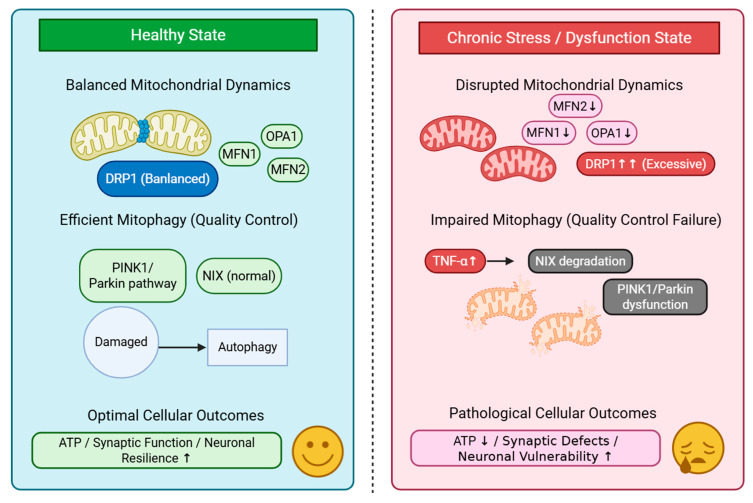
Schematic illustration of mitochondrial dynamics and mitophagy in health and chronic stress-induced dysfunction leading to mental health implications.

**Table 1 ijms-26-07611-t001:** Key biomarkers linking metabolism and mental health.

Biomarker	Origin/Source	Link to Mental Health	Link to Metabolic Health	Potential Clinical Application	Relevant Section(s)
GDF15	Adipose tissue macrophages (stress-induced lipolysis); muscle tissue (mitochondrial stress)	Stress-responsive biomarker, links to anxiety circuits; circadian waking response similar to cortisol; linked to prenatal stress	Linked to peripheral metabolism; Chronic increases linked to weight loss and metabolic issues; levels rise with age, linked to frailty and inflammation	Non-invasive stress monitoring (saliva); early intervention for at-risk individuals	Section 3.1, Section 3.2, Section 3.3, Section 3.5, Section 6.2, Section 7.3, Section 8.1, Section 9.2, Section 10
Ceramides	Gut bacteria (gut dysbiosis); Sphingolipid metabolism	Directly impair hippocampal mitochondrial function; linked to corticosterone-induced depression; linked to prenatal depression	Involved in glycerophospholipid and sphingolipid metabolism	Targeting with probiotics to reduce depressive-like behaviors	Section 2.1, Section 5.2, Section 8.1 and Section 10
Succinate Dehydrogenase (SDH)	Mitochondrial enzyme	Lower serum levels correlated with attenuated frontal-temporal brain activation in MDD patients	Key mitochondrial enzyme	Potential peripheral biomarker of mitochondrial dysfunction in psychiatric disorders	Section 4.4 and Section 7.3
NIX (BNIP3L)	Mitophagy receptor (outer mitochondrial membrane protein)	Degradation linked to accumulation of damaged mitochondria, synaptic defects, and passive stress-coping behaviors in depression models; lower levels in MDD patients; restoration by ketamine and TNF-α blockers reverses behavioral problems	Essential for maintaining neuron health during stress (mitophagy)	Therapeutic target for depression (promoting mitophagy)	Section 4.3 and Section 10
Acylcarnitines	Involved in energy processes	Changes in short-chain acylcarnitine levels relate to presence and severity of depression, especially energy imbalance symptoms; profiles can predict treatment responses	Measures energy capacity; reflect issues with cellular metabolism	Metabolic resilience biomarkers to identify at-risk individuals; guiding treatment options for depression	Section 7.2, Section 8.1 and Section 10
Circulating Mitochondrial DNA (cf-mtDNA)	Mitochondria	Linked to prenatal stress and pregnancy outcomes; potential sign of mitochondrial dysfunction in psychiatric disorders, especially in older adults with mild cognitive impairment and remitted MDD	Mitochondrial health marker	Potential peripheral biomarker of mitochondrial function for brain health and treatment response	Section 3.2 and Section 7.3

**Table 2 ijms-26-07611-t002:** Therapeutic strategies targeting metabolic–psychiatric links.

Therapeutic Approach	Primary Target	Mechanism of Action	Examples/Key Findings	Relevant Section(s)
Mitochondrial Function Enhancement	Brain Mitochondria	Improving mitochondrial electron transport chain activities; restoring mitochondrial function; promoting mitophagy; reducing oxidative stress	Extremely low-frequency electromagnetic field therapy (activates Sirt3-FoxO3a-SOD2 pathway); ketamine (restores respiratory chain activity); natural products (e.g., 20(S)-Protopanaxadiol, Morinda officinalis oligosaccharides, gypenosides, diosmetin); Agomelatine (suppresses hippocampal oxidative stress)	Section 4.5, Section 7.1 and Section 10
Gut Microbiota Modulation	Gut Microbiota	Normalizing gut microbiota composition; lowering ceramide levels; restoring microbial balance; influencing glycerophospholipid metabolism	Probiotics (e.g., Bifidobacterium pseudolongum, Lactobacillus reuteri); synbiotic interventions; dietary changes; fecal microbiota transplantation; traditional medicine compounds (e.g., total flavone of Abelmoschus manihot)	Section 5.3, Section 7.1 and Section 10
Stress Signaling Pathway Targeting (GDF15/GFRAL)	GDF15-GFRAL Pathway	Modulating GDF15 levels or GFRAL receptor activity	Targeting GFRAL for new anxiety treatments; careful consideration of GDF15 metabolic roles (weight loss)	Section 3.5 and Section 10
Lifestyle Interventions	Multiple Pathways (Mitochondria, Neuroplasticity, Overall Health)	Boosting mitochondrial function; improving neuroplasticity; reducing psychiatric disorders; enhancing resistance to inhibitors; regulating BDNF	Physical exercise (aerobic exercise)	Section 4.5 and Section 7.1
Nutritional Approaches	Mitochondrial Function, Neurotransmitter Signaling, Oxidative Damage	Protecting mitochondria and membrane lipids; supporting neurotransmitter signaling	omega-3 fatty acids; antioxidants; vitamin B compounds; magnesium; anti-inflammatory diets	Section 4.5 and Section 7.1
Inflammation Targeting	Inflammatory Pathways	Reducing pro-inflammatory cytokines; addressing neuroinflammation	Theobromine (suppresses neuroinflammation related to nicotine withdrawal); traditional medicine approaches; Infliximab (blocks TNF-α)	Section 4.4 and Section 10
Precision Medicine/Biomarker Guided	Individual Pathways/Patient Profile	Identifying individual metabolic patterns; tailoring therapies; predicting treatment response	Metabolomic profiling (acylcarnitine profiles); multi-omic integration for blood biomarkers; circulating mitochondrial DNA; genetic profiling (mitochondrial SNPs, stress response genes, metabolic capacity)	Section 7.2, Section 7.3, Section 8.1 and Section 10

**Table 3 ijms-26-07611-t003:** Limitations and critical considerations for clinical translation.

Category	Specific Consideration	Description/Challenge	Relevant Section(s)
Translational Challenges	Species differences	Rodent findings may not fully apply to humans due to metabolic, stress response, and brain structure differences. Human mental health is more complex.	Section 9.1
	Variability in compound absorption/processing	Pharmacokinetics can differ significantly between controlled lab conditions and clinical settings.	Section 9.1
	Timing and duration of interventions	Unclear optimal timing, treatment duration, and long-term effects of pathway changes.	Section 9.1
Methodological Considerations	Confounding variables	Diet, physical activity, sleep, medication, and other medical conditions can influence both metabolic and mental health outcomes.	Section 9.2
	Biomarker variability	The dynamic nature of many biomarkers requires clear protocols for sample collection and interpretation.	Section 9.2
	Reproducibility issues	Concerns in metabolomics and microbiome research due to analytical method differences and population variations. Standardization and multi-group validation are crucial.	Section 9.2
Ethical and Safety Considerations	Long-term effects of interventions	Unknown long-term consequences of targeting fundamental metabolic pathways (e.g., GDF15 signaling, mitochondrial function) could lead to unexpected problems.	Section 9.3
	Fair access to personalized therapies	Individual variability in treatment response raises ethical questions about equitable access to tailored interventions.	Section 9.3
	Risks of early/preventive intervention	Weighing benefits of early intervention in high-risk individuals against risks of unnecessary treatment.	Section 9.3

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
