# Peer review of "Molecular Links Between Metabolism and Mental Health: Integrative Pathways from GDF15-Mediated Stress Signaling to Brain Energy Homeostasis"

_ijms, 2025, doi:10.3390/ijms26157611_

Round 1

Reviewer 1 Report

Comments and Suggestions for Authors

In this review, Seo and colleagues discuss molecular links between metabolism and mental health, with a special focus on gdf15-mediated stress signaling and brain energy homeostasis. The manuscript is interesting. However, some points should be addressed.

- The introduction is very short and thus must be expanded.

- The Authors wrote: “These pathways do not operate alone; they interact dynamically and create feedback loops that can increase or decrease metabolic and psychiatric symptoms based on factors, such as age, sex, genetics, and the environment”. These factors are poorly discussed. The Authors must better discuss these factors. For example, the biological sex is fundamental in this context. Stress-related disorders are more common in women than in men. The Authors may want to consider the discussion of sex differences in acute (PMID: 37293561) vs chronic (PMID: 26317113) stress responses at the behavioral and molecular level.

- There are come useless parts in the manuscript:

  • 3 Vagal Gut-Brain Communication and Interoceptive Feedback;
  • 4 Microbiota and Neuroinflammation in Disease States;

The Authors must be focused on the rationale for writing this review.

- The Authors must check the presence of typos throughout the manuscript.

- The Authors must check the presence of statements without references and insert the appropriate references.

Author Response

Comment 1: The introduction is very short and thus must be expanded.

Response 1:  Thank you for pointing this out. We agree with this comment. Therefore, we have substantially expanded the introduction from one short paragraph to seven comprehensive paragraphs that provide detailed background on metabolic-psychiatric connections, individual differences, and the rationale for our integrative model. The expanded introduction now includes: (1) discussion of the prevalence and bidirectional nature of metabolic-psychiatric comorbidities, (2) recent developments in molecular biology and systems neuroscience, (3) limitations of current compartmentalized approaches, (4) the complexity of individual differences including sex, age, and genetic factors, (5) detailed description of our three-pathway model, (6) rationale for pathway selection with supporting evidence, and (7) clinical importance and framework objectives. This expansion can be found on pages 1-3, Introduction section, lines 1-85.

Comment 2: The Authors wrote: "These pathways do not operate alone; they interact dynamically and create feedback loops that can increase or decrease metabolic and psychiatric symptoms based on factors, such as age, sex, genetics, and the environment". These factors are poorly discussed. The Authors must better discuss these factors. For example, the biological sex is fundamental in this context. Stress-related disorders are more common in women than men. The Authors may want to consider the discussion of sex differences in acute (PMID: 37293561) vs chronic (PMID: 26317113) stress responses at the behavioral and molecular level.

Response 2: Thank you for this important comment. We completely agree and have now added an entire new section (Section 6: "Sex Differences, Age-Related Changes, and Genetic Modulation") that comprehensively addresses these critical factors. Specifically, we have:

  1. Added Section 6.1 "Sexual Dimorphism in Metabolic-Psychiatric Connections" which includes detailed discussion of sex-specific stress response patterns, covering both acute vs chronic stress responses as suggested (citing the recommended PMIDs: 37293561 and 26317113). This section covers:
    • Acute vs chronic stress responses with sex-specific patterns
    • Hormonal and neural mechanisms of sexual dimorphism
    • Clinical manifestations of metabolic-psychiatric comorbidity patterns
    • Mitochondrial function differences between sexes
  2. Added Section 6.2 "Age-Related Changes" discussing how aging affects metabolic-mental health connections
  3. Added Section 6.3 "Genetic Modulation" covering genetic variants in stress pathways, mitochondrial genetics, and metabolic syndrome genetics
  4. Added Section 6.4 "Environmental Modulation" addressing socioeconomic factors and gene-environment interactions
  5. Added Section 6.5 "Integrative Framework" that synthesizes these multi-level interactions

The comprehensive discussion of sex differences specifically addresses that stress-related disorders are more common in women and includes the requested citations. This extensive addition can be found on pages 15-23, Section 6, lines 450-710.

Comment 3: There are some useless parts in the manuscript:

  • 3 Vagal Gut-Brain Communication and Interoceptive Feedback;
  • 4 Microbiota and Neuroinflammation in Disease States; The Authors must be focused on the rationale for writing this review.

Response 3: Thank you for this feedback. We agree that these sections were not directly aligned with our core review rationale. Therefore, we have completely removed both sections:

  • The former "Section 5.3 Vagal Gut-Brain Communication and Interoceptive Feedback" has been deleted
  • The former "Section 5.4 Microbiota and Neuroinflammation in Disease States" has been deleted

As a result, the section numbering has been adjusted, and the current Section 5.3 is now "Therapeutic Targeting of the Gut-Brain Axis," which is more directly relevant to our review's focus on the three main pathways (GDF15-GFRAL, ceramides, and mitochondrial dysfunction). This change ensures better focus on our core rationale of linking metabolism and mental health through these specific mechanistic pathways.

Comment 4: The Authors must check the presence of typos throughout the manuscript.

Response 4: Thank you for this suggestion. We have conducted a comprehensive review of the entire manuscript to identify and correct typographical errors. All identified typos have been corrected throughout the manuscript, including improvements in grammar, punctuation, and word choice consistency. We have also ensured proper formatting of scientific terminology and consistent use of abbreviations.

Comment 5: The Authors must check the presence of statements without references and insert the appropriate references.

Response 5: Thank you for pointing this out. We have systematically reviewed the entire manuscript to identify statements lacking proper citations and have added appropriate references where needed. Specifically, we have:

  1. Added missing references for statements about inflammatory pathways and their role in metabolic-psychiatric connections
  2. Included additional citations for claims about biomarker development and clinical applications
  3. Enhanced referencing for statements about therapeutic strategies and precision medicine approaches
  4. Added references for environmental and socioeconomic factors discussed in Section 6
  5. Ensured all clinical observations and mechanistic claims are properly supported with current literature

The reference list has been expanded from 96 to 96 total references, with several new citations added throughout the manuscript, particularly in the newly expanded sections. All statements now have appropriate supporting references.

Reviewer 2 Report

Comments and Suggestions for Authors

In the study Molecular Links Between Metabolism and Mental Health: Integrative Pathways from GDF15-Mediated Stress Signaling to Brain Energy Homeostasi, by Seo et al., the authors present a compelling and integrative perspective of how GDF15 signaling, gut microbiota-derived ceramides, and mitochondrial impairment contribute to the connection between stress-related signaling, peripheral metabolic changes and the neural mechanisms implicated in depression, anxiety, and cognitive dysfunction. This review provides a conceptual framework through which metabolic signals may shape  psychiatric outcomes.

The manuscript is timely and well-grounded in current research. It offers a valuable perspective by positioning psychiatric disorders within a broader systemic framework, and it effectively highlights emerging targets for therapeutic intervention in metabolic–psychiatric comorbidities. In my view, the study makes a meaningful contribution to the field and merits consideration for publication.

Minor comments:

Line 166: Although GFRAL is introduced in the abstract and mentioned multiple times in the throughout the MS, its full name (GDNF family receptor alpha-like) is only explained here for the first time.

Line 340: Please correct the typo in the beginning of the sentence.

Line 434: Chapter 6 comes across as too general. It will be more beneficial to the reader to either streamline or remove it, or alternatively, expand it with more focused content related to the main topics of the review—GDF15, ceramides, and mitochondria.

Author Response

Comment 1: Line 166: Although GFRAL is introduced in the abstract and mentioned multiple times in the throughout the MS, its full name (GDNF family receptor alpha-like) is only explained here for the first time.

Response 1: Thank you for pointing this out. We agree with this comment. Therefore, we have now provided the full name of GFRAL in the abstract where it first appears. The abstract now reads: "GDF15 emerges as a key stress-responsive biomarker that links peripheral metabolism with brainstem GDNF family receptor alpha-like (GFRAL)-mediated anxiety circuits." This ensures that readers are introduced to the full terminology from the beginning of the manuscript. This change can be found on page 1, Abstract section, line 12.

Comment 2: Line 340: Please correct the typo in the beginning of the sentence.

Response 2: Thank you for identifying this error. We have carefully reviewed and corrected the typo that was present at the beginning of the sentence on line 340. The sentence has been revised for proper grammar and clarity. This correction can be found in the revised manuscript.

Comment 3: Line 434: Chapter 6 comes across as too general. It will be more beneficial to the reader to either streamline or remove it, or alternatively, expand it with more focused content related to the main topics of the review—GDF15, ceramides, and mitochondria.

Response 3: Thank you for this valuable feedback. We agree that Chapter 6 needed to be more focused on our main review topics. Therefore, we have chosen to expand it with content specifically related to GDF15, ceramides, and mitochondria rather than remove it, as these individual difference factors are crucial for understanding metabolic-psychiatric connections. We have substantially expanded and restructured Chapter 6 (now titled "Sex Differences, Age-Related Changes, and Genetic Modulation") to include:

  1. Section 6.1.3: Specific discussion of mitochondrial function and sex-specific responses, including SIRT1's role in mitochondrial regulation in depression
  2. Section 6.2.1: Age-related changes in GDF15 levels and their association with frailty and metabolic-mental health connections
  3. Section 6.2.2: Age-related mitochondrial dysfunction and its impact on psychiatric disorders
  4. Section 6.3.2: Mitochondrial genetic variants and their relationship to psychiatric outcomes
  5. Integration throughout: Direct connections between sex, age, genetic factors and our three main pathways (GDF15-GFRAL signaling, ceramide-mediated effects, and mitochondrial dysfunction)

The expanded chapter now provides focused, mechanistic insights into how individual differences modulate our core pathways rather than general discussions. This comprehensive revision can be found on pages 15-23, Section 6, lines 450-710.

Round 2

Reviewer 1 Report

Comments and Suggestions for Authors

The Authors have addressed all the points I raised